# Differential Pd-nanocrystal facets demonstrate distinct antibacterial activity against Gram-positive and Gram-negative bacteria

Ge Fang [1], Weifeng Li [1], Xiaomei Shen[2], Jose Manuel Perez-Aguilar [3], Yu Chong[1], Xingfa Gao[2], Zhifang Chai[1], Chunying Chen [4], Cuicui Ge[1] & Ruhong Zhou [1,3,5]

Noble metal-based nanomaterials have shown promise as potential enzyme mimetics, but the facet effect and underlying molecular mechanisms are largely unknown. Herein, with a combined experimental and theoretical approach, we unveil that palladium (Pd) nanocrystals exhibit facet-dependent oxidase and peroxidase-like activities that endow them with excellent antibacterial properties via generation of reactive oxygen species. The antibacterial efficiency of Pd nanocrystals against Gram-positive bacteria is consistent with the extent of their enzyme-like activity, that is {100}-faceted Pd cubes with higher activities kill bacteria more effectively than {111}-faceted Pd octahedrons. Surprisingly, a reverse trend of antibacterial activity is observed against Gram-negative bacteria, with Pd octahedrons displaying stronger penetration into bacterial membranes than Pd nanocubes, thereby exerting higher anti-bacterial activity than the latter. Our findings provide a deeper understanding of facet-dependent enzyme-like activities and might advance the development of noble metal-based nanomaterials with both enhanced and targeted antibacterial activities.

[1] School for Radiological and Interdisciplinary Sciences (RAD-X), Collaborative Innovation Center of Radiation Medicine of Jiangsu Higher Education Institutions, Soochow University, Suzhou, 215123, China. [2] College of Chemistry and Chemical Engineering, Jiangxi Normal University, Nanchang, 330022, China. [3] IBM Thomas J. Watson Research Center, Yorktown Heights, New York 10598, USA. [4] Key Laboratory for Biomedical Effects of Nanomaterials and Nanosafety and CAS Center for Excellence in Nanoscience, National Center for Nanoscience and Technology of China and Institute of High Energy Physics, Chinese Academy of Sciences, Beijing, 100190, China. [5] Department of Chemistry, Columbia University, New York, New York 10027, USA. Ge Fang and Weifeng Li contributed equally to this work. Correspondence and requests for materials should be addressed to C.C. (email: chenchy@nanoctr.cn) or to C.G. (email: ccge@suda.edu.cn) or to R.Z. (email: ruhongz@us.ibm.com)

nfectious diseases induced by bacteria are considered one of the greatest health challenges worldwide, afflicting millions of people annually[1–3]. Not only the emergence of new infectious diseases but also the continuous increase in bacterial drug resistance present serious problems for the preservation of public health. In this regard, engineered nanostructures have emerged as one of the most promising antibacterial agents due to their high surface-to-volume ratio and their exceptional—either intrinsic or chemically incorporated—antibacterial activity[4–6].

With the recent advances in nanoscience and nanotechnology, there is an increasing interest in developing nanomaterials with outstanding antibacterial properties. In particular, nanomaterials, including metals[7, 8], metal oxides[9–11], and carbon-based nanostructures[12–14], have been proposed as potential candidates to overcome the drawbacks of small molecule antibiotics, e.g., drug resistance. However, although elemental nanomaterials, such as silver nanoparticles (NPs), possess an intrinsic broad-spectrum antibacterial activity[15–18], they also present a serious threat due to their toxicity to human cells[19, 20]. Therefore, it is fundamentally important to investigate NPs with intrinsically high antibacterial properties but with a proper human biocompatibility, as viable candidates for safer NP-based antibiotics.

Previous studies have revealed the antibacterial properties of noble metal-based NPs composed of less toxic elements including gold (Au)[21, 22], platinum (Pt)[23, 24], and palladium (Pd)[25, 26]. For instance, Huo et al.[27] found that zwitterionic Au NPs displayed effective antibacterial activity by controlling their size and surface charge orientation without significant toxicity to human cells. Moreover, Zhao et al.[28] showed that biometallic Au/Pt NPs exhibited high antibacterial potency even without any surface modification. Evidence has emerged suggesting that the antibacterial properties of noble metal-based NPs are usually attributed to their oxidase- and peroxidase-like activities[29].

It is also worth noticing that the surface morphology of NP has an important role in regulating their various catalytic activities[30–32] and thus the enzyme-mimetic activity of noble metal-based NPs could be controlled by tuning their exposed facets. Along these lines, Long et al.[33] have demonstrated that surface facet is a key parameter to modulate the $O_2$ activation process on metal nanocrystals. Therefore, it is highly anticipated that the antibacterial activities of noble metal-based NPs could be regulated by simply tuning their exposed facets.

Here we report the synthesis of Pd nanocrystals with different exposed facets to tune their enzyme-like activities and antibacterial behavior against both Gram-positive and Gram-negative bacteria. We find that Pd nanocrystals not only display substantial facet-dependent antibacterial activities, but also exhibit a surprising reverse antibacterial mode of action between Gram-positive and Gram-negative bacteria. Using a combination of experimental and theoretical approaches, we delineate possible mechanistic bases for these experimental observations.

## Results

**Preparation and characterization of the Pd nanocrystals.** The synthesis of Pd nanocrystals with the {111} and {100} surfaces were carried out by a hydrothermal method as described in previous works[33]. Transmission electron microscopy (TEM) and high-resolution TEM (HRTEM) images of the samples indicate that the synthesized Pd nanocrystals displayed cubic and octahedral geometries with an average edge length of around 10 nm (Fig. 1a, b). The HRTEM images, corresponding to Fast Fourier Transform (FFT) patterns taken from individual nanocrystals and X-ray diffraction patterns, confirmed that the Pd cubes and Pd octahedrons exhibited the expected {100} and {111} facets, respectively (Fig. 1c–f and Supplementary Fig. 1a). Besides,

Fourier transformation infrared spectra (Supplementary Fig. 1b) combined with X-ray photoelectron spectroscopy (Supplementary Fig. 1d and e) analyses indicated that the stabilizer agent poly (vinyl pyrrolidone) (PVP) is present on the surface of the Pd nanocrystals. Furthermore, the thermogravimetric (TG) analysis showed that there is no obvious weight-loss stage at temperatures up to 800 °C in the TG curves of the Pd nanocrystals under inert conditions (nitrogen), indicating that only small amounts of PVP are present on the surface of nanocrystals (Supplementary Fig. 1c). Overall, these analyses of Pd NP's surfaces show that the differences between various Pd nanocrystals are mainly attributed to the surface facets.

**Oxidase-like and peroxidase-like properties of Pd nanocrystals.** To monitor the oxidase-like property of the Pd nanocrystals, we utilized the 3,3,5,5-tetramethylbenzidine (TMB) molecular probe, which displays a characteristic absorbance at 652 nm when it is oxidized. The UV-Vis spectroscopy measurements show time-dependent increase in TMB oxidation catalyzed by Pd nanocrystals (Supplementary Fig. 2a, b). Interestingly, the Pd cubes more effectively promote TMB oxidation than the Pd octahedrons do (Supplementary Fig. 2c) and the catalytic oxidation rate increased with increasing concentrations of the Pd nanocrystals (Supplementary Fig. 2d).

The oxidase-like property of the Pd nanocrystals was further evaluated by using the physiologically relevant antioxidant, ascorbic acid (AA, also known as Vitamin C). The AA biomolecule is an effective antioxidant agent that has an important role in cellular cyclical redox reactions[34, 35]. As shown in Fig. 2a, the addition of the Pd cubes clearly promotes the AA oxidation, as evidenced by the time-dependent reduction in the maximum absorbance. Consistently, the Pd octahedrons displayed a similar oxidation profile than Pd cubes did but exhibited a weaker capacity to oxidize AA (Fig. 2b). Subsequently, the product of catalytic oxidation process, hydrogen peroxide ($H_2O_2$), was detected. Compared to control (AA alone), a marked increase of $H_2O_2$ are detected in the presence of Pd nanocrystals. Notably, much more significant amount of $H_2O_2$ were generated upon the addition of the Pd cubes than in the case of the Pd octahedrons (Fig. 2c). These results verify that the Pd nanocrystals are capable of catalyzing the oxidation of biologically relevant antioxidant agents, resulting in the production of $H_2O_2$. In general, the superoxide radical is precursor of the formation of $H_2O_2$

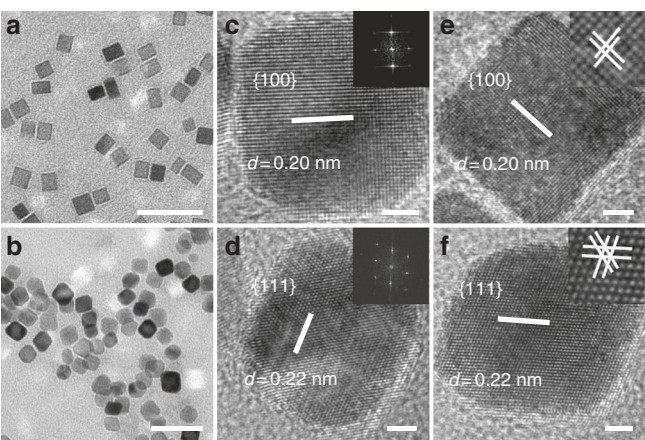

**Fig. 1** Characterization of the Pd nanocrystals. The TEM (**a**, **b**, scale bar: 20 nm), HRTEM (**c–f**, scale bar: 2 nm) and corresponding FFT patterns (inset) of the Pd nanocrystals. The images in **a**, **c**, **e** correspond to the {100}-faceted Pd cubes, whereas those in the **b**, **d**, **f** correspond to the {111}-faceted Pd octahedrons

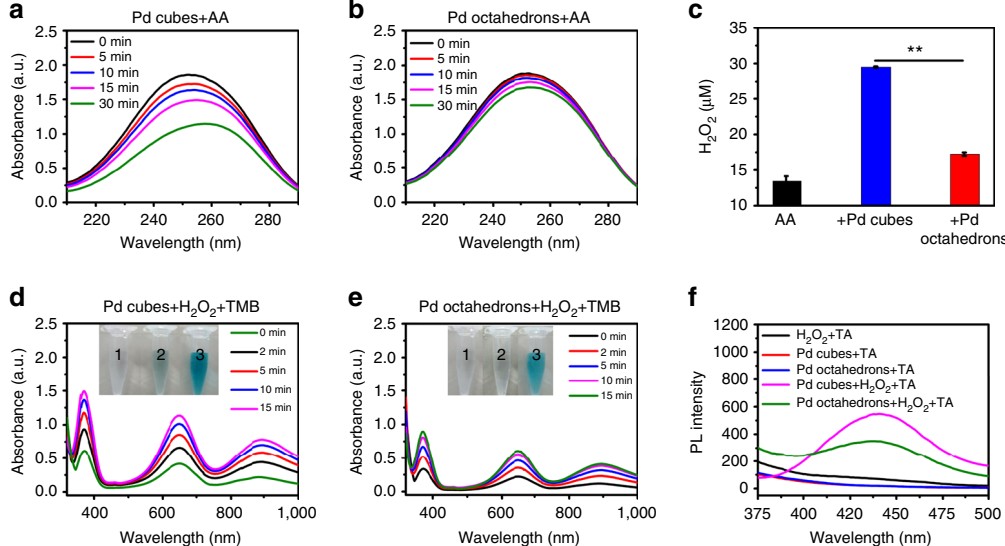

**Fig. 2** Enzyme-like properties of the Pd nanocrystals. Oxidase-like properties of the Pd nanocrystals. Time-dependent absorbance spectra of the AA species catalyzed by **a** the Pd cubes and **b** the Pd octahedrons with the characteristic absorption maximum of AA at 250 nm. **c** The concentration of $H_2O_2$ generated in the catalytic system: AA alone and AA with the different Pd nanocrystals. The data indicate the means and SD from three parallel experiments. *P*-values were calculated by the Student's test: **$p < 0.01$. Peroxidase-like properties of the Pd nanocrystals. Time-dependent absorbance spectra of TMB in different reaction systems: **d** Pd cubes + $H_2O_2$ + TMB and **e** Pd octahedrons + $H_2O_2$ + TMB, where the maximum absorbance values for the TMB$^+$ intermediate (responsible for the characteristic blue color) are at 370 nm and 652 nm, respectively. The inserted images (tubes) represent the visual color changes of TMB in different reaction systems: 1, $H_2O_2$ + TMB; 2, Pd nanocrystals + TMB; 3, Pd nanocrystals + $H_2O_2$ + TMB. **f** The fluorescence spectra for detection of hydroxyl radicals from the different reaction systems. Here, the intermediate fluorescence product, 2-hydroxy terephthalic acid (TAOH), presents a maximum emission wavelength at 435 nm

molecules. We used hydroethidine (HE) as a fluorescence probe for tracking the presence of the superoxide radical. Compared with the control experiment, an obvious increase in the fluorescence intensity was observed upon the addition of the Pd nanocrystals, indicating the generation of superoxide radical species (Supplementary Fig. 3). As expected, this analysis corroborates the fact that the Pd cubes can more efficiently promote the generation of superoxide radical than the Pd octahedrons.

We investigated the peroxidase-like property of the Pd nanocrystals by following the catalysis of the peroxidase substrate TMB by $H_2O_2$. In the redox mechanism of TMB, the substrate is oxidized to form the one-electron oxidation intermediate TMB$^+$. The results from the time-dependent absorbance spectra (Fig. 2d, e) imply that the Pd nanocrystals are able to catalyze the oxidation of TMB via $H_2O_2$, as the expected intermediate product was detected (inset in Figs. 2d, e), which in turn is an indication of the NP's peroxidase-like property. Moreover, the Pd nanocrystals also exhibit facet-dependent peroxidase-like property with the cubic nanocrystals displaying higher activities than the octahedral counterparts.

One possible mechanism for the peroxidase-like property of the Pd nanocrystals might originate from the nanocrystal's ability to decompose $H_2O_2$ species to generate hydroxyl radicals (•OH). Therefore, we used terephthalic acid (TA) to monitor the formation of •OH, which would react with TA to form a highly fluorescence product, 2-hydroxy TA (TAOH)[36]. Accordingly, fluorescence experiments were carried out and as shown in Fig. 2f and Supplementary Fig. 4, intense fluorescence was detected in the system containing TA, $H_2O_2$, and Pd nanocrystals, whereas only a very weak fluorescence signal was detected in the absence of either $H_2O_2$ or Pd nanocrystals. This observation proves that the •OH species are mainly generated from the decomposition of $H_2O_2$ catalyzed by the Pd nanocrystals. Notably, the fluorescence intensity produced by the Pd cubes is significantly stronger than

that of the Pd octahedrons (**$p < 0.01$), indicating that the {100}-faceted Pd cubes more efficiently catalyze the generation of •OH radicals from $H_2O_2$.

**Quantum mechanics calculations on the catalytic activities**. To provide detailed insights into the distinct catalytic activities of the Pd nanocrystals, we performed spin-polarized density functional theory (DFT) calculations based on quantum mechanics to estimate the energetics involved during the oxidase- and peroxidase-like processes. First, we calculated the dissociation energy barriers for a system composed of one $O_2$ molecule on each of the different Pd surfaces. In previous studies, we have reported[37] that the oxidase-like activity of a metal is closely related to the dissociative adsorption of the $O_2$ species on the respective metal surface (Eq. 1):

$$O_2 = 2 \bullet O \qquad (1)$$

According to this mechanism, the Pd surface catalyzes the dissociation of the $O_2$ molecule to yield single O atoms, which are responsible for the subsequent oxidation of organic substrates. For the surface-bound $O_2$ dissociation process, the magnitude of oxidase-like activity can be estimated by the activation energy ($E_a$) of the process illustrated in Eq. 1, where a lower $E_a$ indicates a higher oxidase-like activity. The dissociative energy profiles for the $O_2$ molecule on the Pd {111} and {100} surfaces are summarized in Fig. 3a, b, respectively. The $O_2$ binding on the Pd {111} facet has an adsorption energy of −0.85 eV, whereas for the $O_2$ on the Pd {100} facet, the value is −1.40 eV. These results indicate that the Pd {100} facet, present in the Pd cubes, exhibits a stronger attraction for the $O_2$ molecule. The activation energies ($E_a$) of the surface-bound $O_2$ dissociation for the {111} and {100} facets were also calculated with values of 0.67 and 0.31 eV, respectively. From these results, we conclude that the dissociative adsorption of the $O_2$ molecule on the Pd {100} facet is

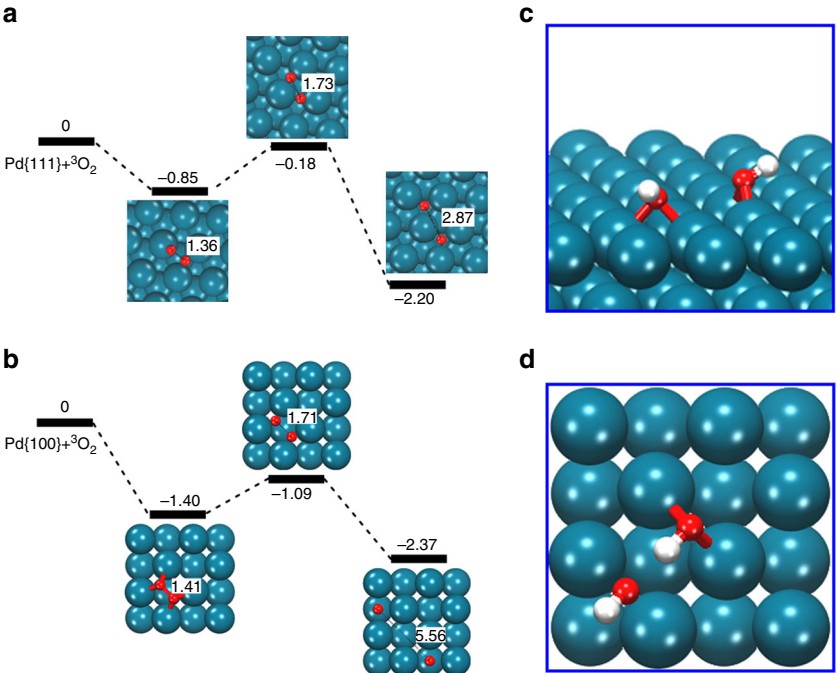

**Fig. 3** Energy profiles of $O_2$ dissociative adsorption on Pd surfaces. Relative energies (eV) and O–O atomic distances (Å) are depicted (**a** and **b**). The lowest-energy adsorption structures for the two oxygen-containing species (•OH) on the Pd{111} and Pd{100} facets (**c** and **d**)

energetically more favorable than that on the Pd {111} facet, corresponding to a higher oxidase-like activity by the former.

The peroxidase-like activity on a metal surface under acidic conditions involves the two-step process described in equations 2 and 3. In the first step, the adsorbed $H_2O_2$ species dissociates to generate •OH radicals (Eq. 2), which subsequently rearrange to yield the adsorbed $H_2O$ and the adatom O (Eq. 3).

$$H_2O_2 = 2 \bullet OH \qquad (2)$$

$$2 \bullet OH = H_2O + O \qquad (3)$$

Similar to the case of the oxidase-like activity, the resulted O adatom is responsible for the oxidation of organic substrates. From previous studies, we found that the reaction depicted in Eq. 2 is the rate-determining step, whose overall reaction energy ($E_r$) can be used as a descriptor of the peroxidase-like activity— i.e., a more negative value of $E_r$ indicates a higher peroxidase-like activity[37]. The lowest-energy adsorption structures for the two •OH species, generated by the homolytic cleavage of $H_2O_2$ on the Pd {111} and {100} facets, are depicted in Fig. 3c, d, respectively. For these •OH structures, the $E_r$ values for the Pd {111} and {100} facets were calculated to be − 2.26 and − 2.91 eV, respectively. These values indicate that, on the Pd {100} facet, the homolytic dissociation of the adsorbed $H_2O_2$ entity is more favorable than on the Pd {111} facet, which suggests a stronger peroxidase-like activity by the former. This result is consistent to the aforementioned higher oxidase-like activity of the Pd {100} facet.

**Antibacterial activity of the Pd nanocrystals.** We have revealed that the Pd nanocrystals exhibit outstanding oxidase- and peroxidase-like properties by efficiently generating $H_2O_2$ species that are subsequently converted into •OH radicals. As Pd nanocrystals are highly expected to kill bacteria by inducing the formation of reactive oxygen species (ROS), we evaluated the respective antibacterial activities for both, Pd cubes and Pd octahedrons. As shown in Fig. 4a, Pd nanocrystals efficiently

inhibited the proliferation of Gram-positive bacteria—the drug-resistant *Staphylococcus aureus*. Notably, a facet-dependent antibacterial activity is observed, where the Pd cubic NPs were able to kill more efficiently the Gram-positive *S. aureus* than the Pd octahedral counterparts.

In order to observe the morphological changes in *S. aureus* induced by the Pd NPs, scanning electron microscopy (SEM) was used. In the absence of Pd NPs (control group), the *S. aureus* cells display spherical and smooth structures, whereas after the treatment with Pd octahedrons, the cells became disrupted with irregular and wrinkled morphology. Remarkably, after the treatment with Pd cubes, the bacterial cells are found to be more seriously damaged showing a complete loss of the membrane integrity (Fig. 4b). These morphological changes not only corroborate the beneficial antibacterial activities of the Pd nanocrystals against Gram-positive bacteria but also demonstrate the superior properties of Pd cubes to efficiently kill the drug-resistant *S. aureus*.

To further examine the antibacterial activities of the Pd nanocrystals, fluorescence-based Live/Dead assays were performed. Most bacteria in the control group remained alive, as indicated by the lack of red fluorescence (Fig. 4c). Addition of the Pd octahedrons resulted in a perceptible increase in the number of dead cells, which was evidenced by the intensification of the red fluorescence signal. Remarkably, treatment with the Pd cubes caused a dramatic increase in the number of dead cells, as indicated by the dominant red fluorescence signal. These results further substantiate the facet-dependent antibacterial activity of Pd nanocrystals with a superior efficacy by the Pd cubes.

To provide detailed mechanistic insights into the antibacterial activity of Pd NPs, dissolution experiments were carried out and the results showed that extremely small amounts of $Pd^{2+}$ are released from the Pd nanocrystals (Supplementary Fig. 5). The morphological features of the different Pd nanocrystal structures, inside or outside of the bacterial cells, were characterized by TEM images and no meaningful changes were observed for any of the Pd nanocrystals, either inside or outside of the bacteria, indicating

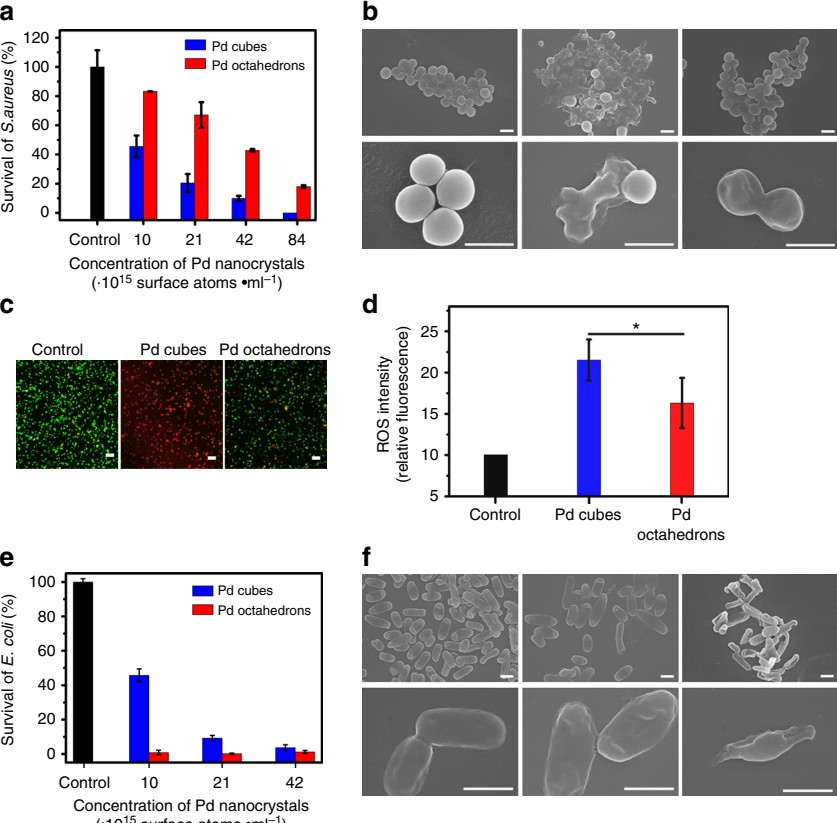

**Fig. 4** Distinct antibacterial activity of Pd nanocrystals. Both Gram-positive and Gram-negative bacteria are tested for the antibacterial activity with Pd nanocrystals. **a** Survival rates of *S. aureus* treated with various concentrations of Pd nanocrystals in terms of the number of surface atoms per milliliter. The colony-forming units counting method was applied to evaluate the actual antibacterial effects of the Pd nanocrystals. The data indicate the means and SD from three parallel experiments. **b** Typical SEM images of the *S. aureus* cells exposed to various treatments. Scale bar: 1 μm. **c** Representative fluorescence images of live (green) and dead (red) cells. Scale bar: 10 μm. **d** Quantitative analysis of the ROS levels in *S. aureus* cells after different treatments. The data indicate the mean fluorescence intensity and SD from three regions. *P*-values were calculated by the student's test: *$p < 0.05$. **e** Survival rates of *E. coli* treated with various concentrations of the Pd nanocrystals in terms of the number of surface atoms per milliliter. The data indicate the means and SD from three parallel experiments. **f** Typical SEM images of *E. coli* cells exposed to various treatments. Scale bar: 1 μm

robust NP stability (Supplementary Fig. 6). Thus, these results seem to indicate that even though there is release of trace amounts of $Pd^{2+}$ from the Pd nanocrystals, their respective geometric characteristics remain robust under intracellular and extracellular conditions.

Next we determined the intracellular levels of ROS to examine whether the bacterial death is caused by oxidative damage (oxidative stress). We found that, relative to the control group, the peroxide levels remarkably increase upon exposed to Pd nanocrystals, indicated by the clear fluorescence signal (Supplementary Fig. 7), and treatment with the Pd cubes significantly induced higher levels of ROS than the Pd octahedrons (Fig. 4d). Our results suggest that the antibacterial property is caused by the generation of ROS catalyzed by the Pd nanocrystals. Moreover, the antibacterial activities exhibit a facet dependence, i.e., the Pd {100} facet bears higher antibacterial activities than the {111}-facet, which is in agreement with their respective enzyme-like activities. Furthermore, we used HE and 3'-p-(hydroxyphenyl) fluorescein as fluorescent probes to respectively measure the formation of superoxide and hydroxyl radicals inside the bacterial cells. Relative to the control group, a strong fluorescence signal was observed upon addition of Pd nanocrystals, indicating the generation of hydroxyl and superoxide radicals (Supplementary Fig. 8). Notably, treatment with the Pd cubes significantly induced higher levels of hydroxyl radicals than the Pd octahedrons (Supplementary Fig. 8a, c).

We further extended our characterization of the antibacterial activities of Pd nanocrystal by including the Gram-negative bacteria *Escherichia coli*. It was surprising to observe a reverse trend, as the Pd octahedrons are more effective in killing *E. coli* cells than the Pd cubes (see Fig. 4e). The untreated *E. coli* cells display a typical rod-shaped structure with smooth and intact cell walls, whereas exposure to Pd cubes causes their morphology to become uneven and wrinkled (Fig. 4f). Notably, treatment with Pd octahedrons induces cell lysis with a complete loss of cellular integrity. The Live/Dead assays also provide a consistent finding that, relative to the limited killing efficiency of the Pd cubes, the Pd octahedrons exhibit an outstanding killing potency against *E. coli* (Supplementary Fig. 9a). Furthermore, from the magnitude of the intracellular ROS (Supplementary Fig. 9b, c), higher peroxide levels are observed in the presence of the Pd octahedrons than in the Pd cubes. In addition, treatment with the Pd octahedrons significantly induced higher levels of hydroxyl and superoxide radicals than those induced by the Pd cubes (Supplementary Fig. 8b, d).

In order to further confirm the reversed antibacterial trend, we fabricated Pd tetrahedrons with the same {111} facets and similar size to the above Pd octahedrons, and investigated the antibacterial activity of these Pd tetrahedrons against *E. coli*. We found that {111}-faceted Pd tetrahedrons display higher antibacterial activity than {100}-faceted Pd cubes (Supplementary Fig. 10), in agreement with the foregoing results. Moreover, the

Pd tetrahedrons showed a little higher antibacterial activity than Pd octahedrons. In essence, these results indicate that the facet and geometric shape co-contribute to the antibacterial activity of Pd nanocrystals, with the shape having a more important role than facet in this case, and thus reversing the trend.

Furthermore, we found that with the increasing size of the Pd nanocrystals[38, 39], the trend of antibacterial activity, previously found for the smaller NPs, remains unchanged (Supplementary Fig. 11), but the difference between the two nanocrystals becomes less profound, indicating that the edge atoms do contribute more than the facial atoms on average. All these observations confirmed that the Pd octahedrons are more effective in killing Gram-negative bacteria than the Pd cubes, which is just the opposite trend observed in Gram-positive bacteria.

In order to further strengthen our conclusion, we also investigated other bacterial species, including *Enterococcus faecalis* (CICC 23658, Gram-positive) and *Salmonella enteritidis* (CICC 21482, Gram-negative). The antibacterial activities of the Pd nanocrystals against these additional bacterial species were evaluated as previously described. The results from these systems displayed similar trends regarding the antibacterial properties of the Pd nanocrystals as those observed for the Gram-positive *S. aureus* and the Gram-negative *E. coli* bacteria (Supplementary Fig. 12).

**Molecular dynamics simulations of the membrane penetration.** Puzzled by the reverse antibacterial mode of action in Gram-negative bacteria, we carried out atomistic molecular dynamics (MD) simulations for the two Pd NPs interacting with an *E. coli*-mimetic membrane. Owing to the structural features of the cell wall in Gram-positive bacteria (thicker layers of peptidoglycan), the cell penetration mechanism for the different Pd NPs was only investigated for the case of Gram-negative bacteria, where meaningful distinct behaviors are expected.

In the simulations, Pd cubic and octahedral nanocrystals of comparable size were initially positioned so that their center of mass (COM) was placed at ~5.5 nm away from the lipid membrane (Fig. 5). To investigate the details of the Pd NP approaching and penetrating the lipid membrane, two different pulling forces ($k_S = 50$ kJ·mol$^{-1}$·nm$^{-1}$ and $k_L = 200$ kJ·mol$^{-1}$·nm$^{-1}$) were introduced to bring the NP from its initial position toward the lipid bilayer (Fig. 5a, b).

For the duration of the Pd cube simulation, we observed that the NP was incapable to completely penetrate the lipid membrane, as one-third of its atoms were still exposed to water. Moreover, during the intrusion event of the Pd cube, the lipid

membrane maintains its structural integrity, as indicated by the lack of structural deformation of its planar morphology. On the contrary, in the case of the Pd octahedron simulations, the NP was able to more easily penetrate the membrane and during the last segment of the simulation, it was able to completely immerse into the lipid bilayer. During the intrusion process of the Pd octahedron, a dented region in the membrane located just below the NP was observed, which indicates the onset of the penetration process. This distinct behavior of the two Pd nanocrystals was quantified by monitoring the vertical separation (along the *z*-coordinate) between the COMs of the Pd particles and the lipid membrane. As depicted in Fig. 5c, the COM of the Pd octahedron approaches faster and reaches deeper into the interior of the membrane than the Pd cubic counterpart (maintaining a difference in the vertical separation of ~0.7 nm). We also characterize the intrusion event by monitoring the time evolution of the interaction energy between each of the Pd NPs and the lipid membrane (Fig. 5d). The interactions are very similar but interestingly, at 100 ns, the interacting curves have not reached a plateau and thus it is expected that given enough time, the penetration process is likely to continue for both NPs.

The results for the simulations with the different pulling forces are in good agreement with each other, exhibiting a remarkable similarity regarding the intrusion features, i.e., the Pd octahedron moves faster and penetrates deeper into the lipid membrane than the Pd cube (Fig. 5 and Supplementary Fig. 13). Interestingly, the simulations with the larger pulling force was employed, our simulations revealed more structural distortions in the membrane caused by the Pd NP intrusion (Supplementary Fig. 13). Although these results should be treated with caution, we noticed that in this case, the interaction between the Pd NPs and lipid membrane is stronger for the case of the Pd octahedron, which would suggest an explanation for the faster and deeper penetration pathway for this particular Pd nanostructure into the *E. coli*-mimetic membrane. Lastly, this simulation reveals a more complete penetration process and perhaps this would elicit more mechanistic distinctions in the interaction between each of the Pd nanocrystals with the bacterial-mimetic lipid membrane.

**Differential cellular internalization of Pd nanocrystals.** Previous studies have found that the cell entry of various NPs, following the membrane penetration mechanism, is highly regulated by the shape, orientation, and surface structure of the particles[40–44]. From the aforementioned external structural differences, it is clear that the two bacterial groups should interact distinctively with the Pd nanocrystals used in this study, prompting us to

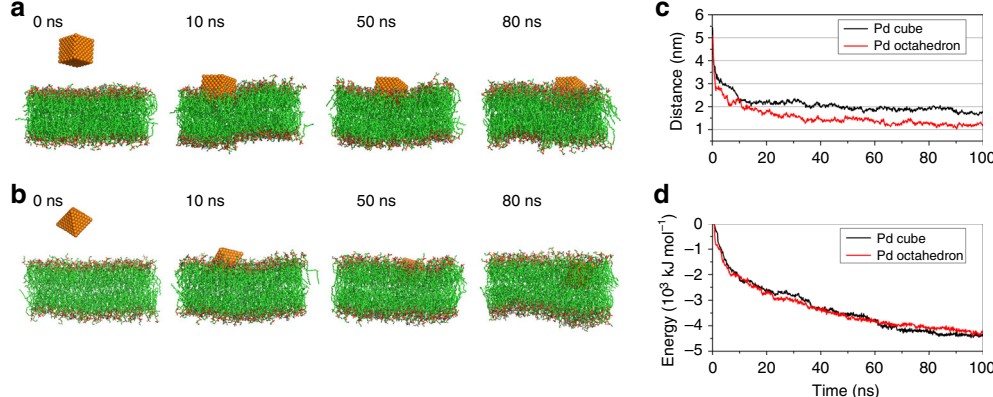

**Fig. 5** Membrane penetration of Pd nanocrystals from simulations. Representative snapshots for the membrane penetration event of **a** the Pd cube and **b** the Pd octahedron. **c** Time evolution of the distance between the center of mass distance of the membrane and the two Pd nanoparticles. **d** Time evolution of the interaction energy between the membrane and the two Pd nanoparticles (here the pulling force constant is $k_S = 50$ kJ·mol$^{-1}$·nm$^{-1}$)

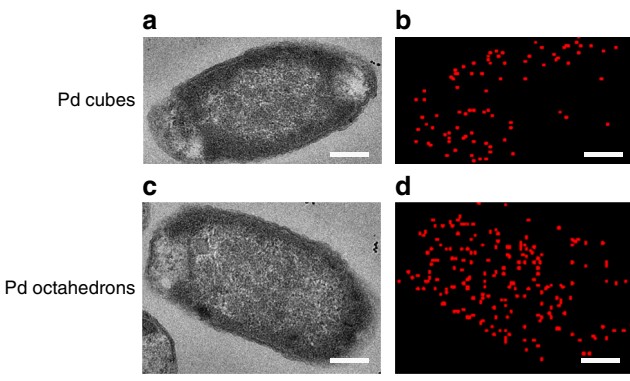

**Fig. 6** The distribution of Pd nanocrystals in *E. coli* cells. TEM images of *E. coli* cells exposed to Pd nanocrystals (**a**, **c**) and HAADF-STEM-EDS elemental mapping images of Pd (**b**, **d**). Scale bar: 200 nm

compare the internalization of the Pd nanocrystals in the two bacterial groups.

From the MD simulation results, the Pd octahedrons would penetrate through the membrane faster and with less effort and thus the Pd octahedrons would be able to accumulate inside Gram-negative bacteria more substantially than the Pd cubes. To support this hypothesis, the distribution of Pd nanocrystals in *E. coli* cells was monitored by scanning TEM-energy dispersive X-ray spectroscopy (EDS) mapping. For *E. coli*, the Pd octahedrons are localized throughout the interior of the cell, whereas the Pd cubes are located mainly at peripheral regions (Fig. 6). These results clearly reflect the differential internalization potentials of the two Pd nanocrystals and are consistent with the observations from the MD simulations. As a result, the highly internalized Pd octahedrons are able to more efficiently deploy their appealing killing effects than the Pd cubes. As for Gram-positive bacteria, *S. aureus*, we found that most of the nanocrystals accumulate on the surface of the cells due to the thick and compact cell membrane (Supplementary Fig. 14). In this case, the antibacterial activities of the two Pd nanocrystals are mainly determined by their respective facet-dependent enzyme-like activities—the more reactive nanostructures, i.e., the Pd cubes, are able to more efficiently elicit their antibacterial effects through the generation of higher ROS levels.

## Discussion

By applying a variety of experimental and theoretical approaches, we found an important correlation between the surface facet of the Pd nanocrystals and their oxidase- and peroxidase-like properties: the {100}-faceted Pd cubes display higher enzyme-like activities than the {111}-faceted Pd octahedrons. The distinctive enzyme-like properties of the Pd nanocrystals endow them with excellent antibacterial properties via the generation of ROS. In Gram-positive bacteria, the Pd nanocrystals mainly accumulate on the surface of cell membrane, thereby exerting antibacterial activities that are in agreement with their enzyme-like activities, i.e., the Pd cubes display a higher potency than Pd octahedrons. However, for Gram-negative bacteria, a reverse trend of the antibacterial activities by Pd nanocrystals was found. Our extensive analyses suggest that the reverse trend is attributed to the differential membrane-penetration capacity of Pd nanocrystals. The Pd cubes exhibit a limited penetration into the bacterial membrane and are mainly localized at peripheral regions, which limits their antibacterial proficiency. On the other hand, the Pd octahedrons show a more efficient translocation into

the interior of bacterial cell, hence fully exerting their antibacterial activity.

Our findings provide the complete description of distinct facet-dependent antibacterial activities of noble metal-based NPs against different bacterial groups, which might open up a new door for developing novel and selective antibacterial applications with these nanomaterials.

## Methods

**Chemical reagents**. Sodium palladium (II) chloride ($Na_2PdCl_4$, ≥ 99.99%), PVP (molecular weight = 55,000 kDa ), L-AA, TMB, citric acid, and TA were obtained from Sigma-Aldrich. KBr, $H_2O_2$, and other reagents were of analytical reagent grade and acquired from Sinopharm Chemical Reagent Co., Ltd. Ultrapure water (18.2 MΩ, Millpore Co., USA) was used throughout the experiment.

**Preparation and characterization of Pd nanocrystals**. The synthesis of Pd nanocrystals with the {111} and {100} surfaces was carried out by a hydrothermal method as described earlier[33]. TEM (Tecnai G-20 spirit BioTwin, FEI, USA) was used to characterize the morphology of Pd nanocrystals and the HRTEM (Tecnai G2 F20 S-TWIN TMP, FEI) images and corresponding FFT patterns were applied for the identification of the surface facets of Pd nanocrystals. More details can be found in Supplementary Methods.

**Oxidase-like property of Pd nanocrystals**. In order to maintain the surface atoms of samples equivalent, different concentrations of Pd nanocrystals dispersions (1.50 mg·ml$^{-1}$ for cubes and 0.54 mg·ml$^{-1}$ for octahedrons) were prepared. For the oxidation of AA, the Pd nanocrystals dispersions were added into 0.1 M AA solution and then UV-Vis absorption spectra were recorded at different times using an UV spectrometer (UV-3600, Shimadzu, Japan).

For the identification of the product of catalytic oxidation, a hydrogen peroxide ($H_2O_2$) assay kit (Beyotime Institute of Biotechnology, Shanghai, China) was employed, which is based on the oxidation of ferrous ions ($Fe^{2+}$) to ferric ions ($Fe^{3+}$) by peroxides and the $Fe^{3+}$ then combine an dye-xylenol orange to form a purple colored complex with the maximum absorbance at 560 nm measurable. In brief, the reaction solution was centrifuged at 14,800 r.p.m. for 15 min and the supernatants were collected. Finally, the surpernatants and detection solution were mixed for 30 min at room temperature and then measured using a microplate reader (BioTek, Synergy NEO, USA). The level of $H_2O_2$ in products was calculated according to a standard concentration curve with triplicate experiments.

**Peroxidase-like property of Pd nanocrystals**. For the oxidation of TMB by $H_2O_2$, the Pd nanocrystals dispersions were diluted with HAc/NaAc buffer (0.2 M : 0.2 M, pH = 4.6) as the substrate and then 10 µl of 100 mM TMB solution and 1 µl of $H_2O_2$ (10 M) was added. Kinetic measurements were carried out by monitoring the absorbance at 652 nm on a UV-Vis spectroscopy (UV-3600, Shimadzu).

For the assay of the product, •OH, TA (hydroxyterephthalic acid) was used as a probe, which could easily react with •OH to form a highly fluorescent product (TAOH). In a typical procedure, TA were added into the reaction solution, and then the fluorescence spectra of the samples were collected using fluorescence spectroscopy (FLS980, Edinburgh).

**Bacterial culture and antibacterial activity test**. *E. coli* (ATCC 6538) and *S. aureus* (ATCC 8739) from China General Microbiological Culture Collection Center were respectively cultured in Luria–Bertani broth medium and Tryptone Soy Broth medium. After reaching the logarithmic phase, the bacteria were diluted to ~ $10^6$ colony-forming units (CFU·ml$^{-1}$) and then supplemented with series concentration of Pd nanocrystals for 20 min. After treatment, the bacterial strains were diluted to $10^3$ CFU·ml$^{-1}$ and cultured on agar plate for 24 h at 37 °C. Finally, the numbers of colonies were counted and the counts on the three plates corresponding to a particular sample were averaged.

**Live/dead fluorescent staining**. After treatment with Pd nanocrystals, both *S. aureus* and *E. coli* bacteria were incubated with SYTO 9 and propidium iodide (Invitrogen Detection Technologies, USA) for 15 min at room temperature in the dark. Then the microscope slide with 5 µl of the stained bacterial suspension was covered with a coverslip and the live/dead bacterial cells were then visualized with confocal laser microscopy (FV1200, OLYMPUS, Japan).

**Preparation of bacterial samples for SEM**. After treatment, bacterial cells were collected with centrifugation at 6,000 r.p.m. for 5 min and fixed with 2.5% glutaraldehyde overnight at 4 °C. After washing with phosphate-buffered saline (PBS) for three times, bacterial cells were dehydrated through graded ethanol solutions. Finally, the obtained samples were put on the silicon glide and observed by SEM (S-4700, Hitachi, Japan).

**Measurement of intracellular ROS**. The oxidant-sensitive dye 2′, 7′-dichlorodihydrofluorescein diacetate (DCFH-DA, Invitrogen Detection Techonologies) was used to measure the intracellular ROS level. After treatment, bacterial cells were stained with 10 µM DCFH-DA for 30 min in the dark at the room temperature and washed them with PBS once. The intracellular ROS levels were measured using confocal laser microscopy (FV1200, OLYMPUS) with excitation wavelength at 488 nm and emission at 525 nm.

**TEM analysis of *E. coli* and *S. aureus* cells**. The distribution of Pd nanocrystals in bacterial cells was evaluated using a combination of TEM and EDS mapping (Tecnai G2 spirit BioTwin, FEI, 120 kV). The samples were prepared for TEM according to the reported method. Briefly, after treatment, the bacterial cells were collected and washed with PBS for three times. In the following, the bacterial cells were fixed with a 2% glutaraldehyde at 4 °C overnight, followed by postfixing for 2 h with 1% osmium tetroxide. The samples were dehydrated with graded ethanol and then infiltrated and embedded in Spurr's resin. Thin sections were mounted on copper grids and then observed on TEM.

**Statistical analysis**. Mean and SD were calculated. Results were expressed as mean ± SD. Comparisons within each group were conducted by a Student's *t*-test.

**DFT calculations of Pd catalytic activity**. Vienna *ab initio* Simulation Package was used to perform spin-polarized DFT calculations, which explicitly considered the atomic spin. Projector augmented wave method was employed to describe the electron–ion interactions. The exchange-correlation functional of Perdew–Burke–Ernzerhof with generalized gradient approximation was used. Geometry optimizations and energy calculations were performed using an energy cut-off with 400 eV and a first order Methfessel–Paxton smearing with 0.2 eV. Four-layered slabs with ($4 \times 4$) and ($2 \times 2$) unit cells in the lateral direction were used to model Pd {111} and {100} surfaces, respectively. A vacuum space of 15 Å was placed above the slabs to avoid mirror interactions. The ($3 \times 3 \times 1$) Monkhorst–Pack mesh *k*-points for ($4 \times 4$) and ($2 \times 2$) unit cells were selected for the calculations. With regard to geometry optimization, the top one layer of the {111} and {100} surfaces was fully relaxed, and the remaining layers were kept fixed. Conjugated-gradient algorithm was employed to optimize the structures. Electronic structures and forces were converged at the criterions of $10^{-6}$ eV and 0.02 eV Å$^{-1}$, respectively

**MD simulations of Pd NPs with lipid membrane**. All the atomistic MD simulations were performed using the GROMACS package. The CHARMM force field was adopted for lipids, and parameters developed by Heinz et al.[45], were adopted for Pd to reproduce the density, surface tension and interface properties with water and bio-organic molecules. The initial structure of the lipid bilayer was generated by the CHARMM-GUI program[46]. Two of the major phospholipids, palmitoyloleoylphosphatidylethanolamine (POPE), and palmitoyloleoylphosphatidylglycerol (POPG), commonly found in Gram-negative bacteria (e.g., *E. coli*) were used in our simulations to model the bacterial cell membrane[47]. In our simulated systems we used a bacterial-mimetic membrane comprised of 258 POPE and 86 POPG lipids in a mixture ratio of 3:1[48–50]. Each system was solvated with ~ 32,500 TIP3P water molecules and sodium ions were added to neutralize the net charge of the entire system; the size of the resulting simulation box is ~ $10.0 \times 10.0 \times 12.8$ nm³. We applied periodic boundary conditions in all directions. The NPT ensemble is used with constant temperature maintained at 300 K by the velocity rescale thermostat with a coupling time constant of 0.1 ps. The *Z*-direction and *X*/*Y* directions are independently coupled at 1 bar by the Berendsen barostat with a time constant of 0.1 ps. The Particle Mesh Ewald method was adopted to treat long-range electrostatic interactions, whereas the vdW interaction was handled with a cutoff distance of 1.2 nm. The bond length for water was constrained with the LINCS algorithm. A time step of 2.0 fs was used and data were collected every 1.0 ps. The simulation system was firstly energy minimized and equilibrated for 10 ns with position restrains applied to the Pd particle and to membrane atoms in accordance with protocols used in our previous studies[51–55], followed by a production simulation phase of 80 ns, where the aforementioned restraints were removed. As documented by the experiments, the intrusion of Pd NP into the bacteria takes more than 20 min to complete, which is inaccessible for the time scale of our MD simulations (time scale: hundreds to thousands of nanoseconds). As a consequence, for data production, biased forces were further applied on the particles to accelerate the occurrence of the intrusion process (see main text).

**Data availability**. The authors declare that the data supporting the findings of this study are available within the paper and the Supplementary Information, or are available from the authors upon request.

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

## Acknowledgements

This work was partially supported by the National Basic Research Program of China (973 Program Grant Number 2014CB931900 and 2016YFA0201600), National Natural Science Foundation of China (11575123, 11574224, and 21320102003), Collaborative Innovation Center of Radiological Medicine of Jiangsu Higher Education Institutions, Jiangsu Provincial Key Laboratory of Radiation Medicine and Protection, and A Project Funded by the Priority Academic Program Development of Jiangsu Higher Education Institutions (PAPD). C.Y.C. appreciates the support from the NSFC Distinguished Young Scholars (11425520).

## Author contributions

The study was planned and directed by C.G., R.Z., C.C., and Z.C. Experiments were conducted by G.F. Y.C. contributed to the improvement of the characterization of the materials. Quantum chemical calculation was conducted by X.S. and X.G. Molecular dynamic simulation was conducted by W.L. The manuscript was prepared by C.G. and R. Z. J.M.P.-A. contributed to the improvement of the manuscript.

## Additional information

**Competing interests:** The authors declare no competing financial interests.

