## [Peer Review File · Nature Communications]

Reviewers' comments:

Reviewer #1 (Remarks to the Author):

This manuscript presents the facet effect of Pd nanocrystals on antibacterial activity. The authors firstly fully describe and discuss the ROS on Pd nanocrystals, and then reveal that Pd nanocubes and octahedrons show different behaviors in the lipid membrane. To support their argument, the authors perform MD simulation and examine the distribution of Pd nanocrystals in cells. This research topic should be of broad interests to the readers; however, more systematic studies should be conducted to fully support the findings in this work. Thus I would like to recommend a major revision before this work is considered for publication in Nature Communications.

1. The authors should perform more control experiments to support their argument. Is the different behaviors of Pd nanocubes and octahedrons in antibacterial activity caused by the proposed facet effect or a geometric shape effect? Which factor plays the crucial role in the membrane penetration of Pd nanocrystals, facet or shape? For instance, the shape of Pd octahedron is covered by {111} facets; other shapes such as tetrahedron also can provide {111} facets. I strongly suggest that the authors perform the control experiments using Pd tetrahedrons and figure out if the geometric shape effect impacts on the membrane penetration.

2. Does particle size affect the behavior of nanocrystals? To establish a reliable structure-property relationship, the research community usually employs the nanocrystals with various sizes to examine the size effect. This effect also has an impact on the facet dependence, as different ratios of face atoms to edge atoms can be achieved when tailoring particle sizes. The authors have to elucidate whether facet atoms or edges atoms make more contributions to the discussed properties when they argue a facet effect.

3. Theoretical simulation is quite important to the findings of this manuscript; however, the simulation results and procedures have not been described in detail. Please include more statements and discussions in the manuscript.

4. The authors should fully characterize the structures of Pd nanocrystals in or out of cells.

Reviewer #2 (Remarks to the Author):

This manuscript present interest information about the correlation between the surface facet of the Pd nanocrystals and their oxidase- and peroxidase-like properties. The manuscript is well written and it has the novelty required to be considered for publication. The conclusions are original and well supported by the results.

Although the experiments and the results are adequate I have the following and only question:

Why surface characterization is not reported (XPS, FTIR, etc)?

It is well known that surface properties of nanocrystals are very important for most of their properties, including antibacterial and oxidase- and peroxidase-like properties. Molecules in the nanocrystal surface could modify their performance.

Authors should demonstrate that the only difference among their Pd nanocrystals is the exposed facet, and the molecules around the surface are the same or they are not present.

Reviewer #3 (Remarks to the Author):

This is high quality research and this reviewer has following comments to improve the manuscript. Results and Discussion: Third paragraph (the last sentence production of H₂O₂) give a reference with an example that catalysis of oxidation of biologically molecules generates H₂O₂. In the next paragraph, authors again make a case of H₂O₂. Generally, O₂⁻ is precursor of the formation of H₂O₂. It would be strengthen to perform such measurements.

A paragraph before Quantum mechanics....: How significant the difference between two intensities, any statistical analysis was done?

Molecular dynamics (MD) simulation... This is too long paragraph or section. Authors need to slowly walk the readers with their thought by dividing this section into 3-4 paragraphs. Please be crisp in writing and to the point.

Differential cellular internalization of the Pd nanocrystals: This reviewer would like to see measurements of ROS in explaining the differences in toxicity. It is okay to describe the results using penetration through membrane, but ROS generated in two facets must differ in giving different antibacterial properties. This will truly strengthen the manuscript. Authors should measure superoxide and hydroxyl radical concentrations in two kinds of facets under antibacterial study of Gram-positive and Gram-negative bacteria. Authors may like to consult the following references on the mechanism part of nanoparticles in general to get across the message out from this very nice piece of work: *Advances Colloid Interface Science* 2015, 225, 229-240; *Advances Colloid Interface Science* 2011, 166, 119-135.

Authors may like to change the title as Differential Pd-Nanocrystal Facets: Distinct Antibacterial Activity against Gram-positive and Gram-negative Bacteria

Reviewer #4 (Remarks to the Author):

The major claims presented in this manuscript are that palladium nanoparticles with different facets, i.e. cubes or octahedrons induce oxidase and peroxidase activity on gram-positive and gram negative bacteria and in doing behave as antibacterial agents. The work includes experimental and theoretical studies to demonstrate the effect of the different facets on the bacteria. In general, the work is interesting, though it does not rise to the level to warrant the novelty or urgency for publication in *Nature Communications*.

The claims of different facets having different effects on gram-positive vs. gram-negative bacteria, while interesting, is quite a stretch when only one species is used, and no experiments are done on other strains. Moreover, the specific strain used for each species is not referred to in the manuscript.

Another important aspect that should be taken into consideration is not only the type of surface but how the stabilizers are organized on the surface. The authors indicate that the stabilizer is PVP, but the role of PVP in these studies has not been considered. Stabilizers play a very important role in how they nanoparticles interact with bacteria and must be taken into consideration. In addition, including the role of the stabilizer in the computational studies would be important is getting a clear picture of the interactions that are being claimed.

It is also important to note that several metal nanoparticles when present in media can slowly undergo

dissolution or other transformations. There is no indication in this manuscript that this was a test conducted. It may be possible that one facet undergoes dissolution differently than another resulting in the formation of ions that will cause toxicity toward the bacteria. Without such tests, the full picture of what may be taking place is missing.

I would suggest revisions and submission to a different journal.

Reviewer #1 (Remarks to the Author):

This manuscript presents the facet effect of Pd nanocrystals on antibacterial activity. The authors firstly fully describe and discuss the ROS on Pd nanocrystals, and then reveal that Pd nanocubes and octahedrons show different behaviors in the lipid membrane. To support their argument, the authors perform MD simulation and examine the distribution of Pd nanocrystals in cells. This research topic should be of broad interests to the readers; however, more systematic studies should be conducted to fully support the findings in this work. Thus I would like to recommend a major revision before this work is considered for publication in Nature Communications.

RESPONSE: We are grateful to the Reviewer for the positive and constructive comments.

1. The authors should perform more control experiments to support their argument. Is the different behaviors of Pd nanocubes and octahedrons in antibacterial activity caused by the proposed facet effect or a geometric shape effect? Which factor plays the crucial role in the membrane penetration of Pd nanocrystals, facet or shape? For instance, the shape of Pd octahedron is covered by {111} facets; other shapes such as tetrahedron also can provide {111} facets. I strongly suggest that the authors perform the control experiments using Pd tetrahedrons and figure out if the geometric shape effect impacts on the membrane penetration.

RESPONSE: We thank the Reviewer for this great comment. As suggested by the referee, we synthesized Pd tetrahedrons (with the same {111} facets) with similar size¹ to the previous as-fabricated Pd octahedrons. Then, we investigated the antibacterial activity of these Pd tetrahedrons against the Gram-negative *E. coli* bacteria. By comparing the antibacterial activity of the Pd tetrahedrons and the Pd cubes, we found agreement with our previous results regarding the distinct and reverse antibacterial activity trend (Figure S10). Some comments about this geometric shape effect were added in page 10, along with the new Figure S10.

In essence, for Gram-positive bacteria, such as *S. aureus*, the thick and compact cell wall is not sensitive to the geometric shape of nanocrystals and thus, the facet plays the dominant role in the antibacterial activities of Pd nanocrystals. On the other hand, for Gram-negative bacteria, such as *E. coli*, the situation is more complicated because the thin cell wall results in differential internalization potentials of Pd nanocrystals with different geometric shapes. Therefore, the facet and geometric shape co-contribute to the antibacterial activity of Pd nanocrystals, and the shape can play a more important role than facet, resulting in a reverse in the trend, for Gram-negative bacteria.

Reference:

[1] Wang Y, Xie S, Liu J, et al. Shape-controlled synthesis of palladium nanocrystals: a mechanistic understanding of the evolution from octahedrons to tetrahedrons. *Nano Lett.* 2013, 13(5): 2276-2281.

2. Does particle size affect the behavior of nanocrystals? To establish a reliable structure-property relationship, the research community usually employs the nanocrystals with various sizes to examine the size effect. This effect also has an impact on the facet dependence, as different ratios of face atoms to edge atoms can be achieved when tailoring particle sizes. The authors have to elucidate whether face atoms or edges atoms make more contributions to the discussed properties when they argue a facet effect.

RESPONSE: Thank you for this concern. To examine the “size effect”, we respectively synthesized larger Pd cubes and Pd octahedrons with sizes around 20 nm^{1, 2}, and then investigated their antibacterial activities against Gram-positive and Gram-negative bacteria. With the increased size of the Pd nanocrystals, the previously observed trend in antibacterial activity remains unchanged. On the other hand, the difference between the two nanocrystals is less accentuated, especially for the case of the Gram-positive bacteria (Figure S11). We observed that for the case of the larger nanoparticles, the Pd octahedrons could more efficiently kill the Gram-negative

bacteria than Pd cubes, whereas the Pd cubes exhibit somewhat higher antibacterial activity than Pd octahedrons against the Gram-positive bacteria. Therefore, with an increasing size of the Pd nanocrystals, the previously observed structure/antibacterial activity relationship is still maintained, but the distinction becomes less profound. As the referee correctly pointed out that the proportion of edge atoms decreases as the particle size increases, our data clearly indicate that the edge atoms do contribute more significantly to the aforementioned properties than those facial atoms.

Some comments about this size effect were added in page 11, along with the new Figure S10.

References:

[1] Jin M, Liu H, Zhang H, et al. Synthesis of Pd nanocrystals enclosed by {100} facets and with sizes < 10 nm for application in CO oxidation. *Nano Res.* 2011, 4(1): 83-91.

[2] Jin M, Zhang H, Xie Z, et al. Palladium nanocrystals enclosed by {100} and {111} facets in controlled proportions and their catalytic activities for formic acid oxidation[J]. *Energy Environ. Sci.* 2012, 5(4): 6352-6357.

3. Theoretical simulation is quite important to the findings of this manuscript; however, the simulation results and procedures have not been described in detail. Please include more statements and discussions in the manuscript.

RESPONSE: Thank you for this comment. We have modified the “Simulation section” so as to have a clearer and more precise description of our methods and observations. Also, we added more details of our simulation methods and procedures in the Methods section.

4. The authors should fully characterize the structures of Pd nanocrystals in or out of cells.

RESPONSE: We thank the reviewer for this suggestion. The morphological features of the different Pd nanocrystal structures, inside and outside of the bacterial cells, were characterized by TEM. According to the analysis of the TEM images, no changes were observed for any of the Pd nanocrystals either inside or outside of the bacteria, indicating robust stability (Figure S6).

Reviewer #2 (Remarks to the Author):

This manuscript present interest information about the correlation between the surface facet of the Pd nanocrystals and their oxidase- and peroxidase-like properties. The manuscript is well written and it has the novelty required to be considered for publication. The conclusions are original and well supported by the results.

RESPONSE: We are greatly encouraged by the very positive comments of the Reviewer and would like to express our gratitude for the Reviewer's appreciation of our results as well as the critical reading and helpful suggestions.

Although the experiments and the results are adequate I have the following and only question:

1. Why surface characterization is not reported (XPS, FTIR, etc)? It is well known that surface properties of nanocrystals are very important for most of their properties, including antibacterial and oxidase- and peroxidase-like properties. Molecules in the nanocrystal surface could modify their performance.

RESPONSE: We thank the reviewer for this valuable suggestion. X-ray photoelectron spectroscopy (XPS) and Fourier-transform infrared spectroscopy (FTIR) were used to analyze the surface properties of the different Pd nanocrystals, and the results show only the presence of remnants of the PVP stabilizer on the surface of the Pd nanocrystals (Figure S1). More detailed analysis are shown as part of the answer in Question 2 (see below).

2. Authors should demonstrate that the only difference among their Pd nanocrystals is the exposed facet, and the molecules around the surface are the same or they are not present.

RESPONSE: Thank you for your suggestions. In the synthesis process of the different Pd nanocrystals, polyvinylpyrrolidone (PVP) was introduced as a stabilizer whereas ascorbate acid (AA) or citric acid (CA) were employed as reducing reagents. The samples were washed one time with acetone, and two times with ethanol, and redispersed with deionized water after each wash, to remove most of the free PVP and reducing molecules by centrifugation. In the FTIR spectra of the Pd nanocrystals, no stretching vibration absorption of the O-H bond was observed ($3700-3200\text{ cm}^{-1}$), which indicates the successful removal of any trace of AA and CA (Figure S1b).^{1,2} However, the presence of PVP was clearly observed. Furthermore, the thermogravimetric (TG) analysis showed that there is no obvious weight-loss stage at temperatures up to $800\text{ }^{\circ}\text{C}$ in the TG curves of the Pd nanocrystals under inert conditions (nitrogen), indicating that only small amounts of PVP are present on the surface of the nanocrystals (Figure S1c). Overall, the PVP on the surface of Pd nanocrystals is negligible and thus the difference between Pd nanocrystals mainly is attributed to the surface facet.

References:

- [1] Wu K H, Wang Y R, Hwu W H. FTIR and TGA studies of poly (4-vinylpyridine-co-divinylbenzene)-Cu (II) complex. *Polym. Degrad. Stabil*, 2003, 79(2): 195-200.
- [2] Xian J, Hua Q, Jiang Z, et al. Size-dependent interaction of the poly (N-vinyl-2-pyrrolidone) capping ligand with Pd nanocrystals. *Langmuir*, 2012, 28(17): 6736-6741.

Reviewer #3 (Remarks to the Author):

This is high quality research and this reviewer has following comments to improve the manuscript.

RESPONSE: We are very grateful to the Reviewer for the very positive comments and the appreciation of potential clinical significance of our current study.

1. Results and Discussion: Third paragraph (the last sentence production of H₂O₂) give a reference with an example that catalysis of oxidation of biologically molecules generates H₂O₂. In the next paragraph, authors again make a case of H₂O₂. Generally, O₂⁻ is precursor of the formation of H₂O₂. It would be strengthen to perform such measurements.

RESPONSE: We thank the Reviewer for this very useful suggestion. Hydroethidine (HE) was used as a fluorescence probe for tracking the presence of O₂⁻ species. The results showed that, compared to the control experiments, an obvious increase in the fluorescence intensity was observed upon the addition of the Pd nanocrystals, indicating the generation of O₂⁻ radical species (Figure S3). As expected, this analysis corroborates that the Pd cubes more efficiently promote the generation of O₂⁻ radicals than the Pd octahedrons.

2. A paragraph before Quantum mechanics....: How significant the difference between two intensities, any statistical analysis was done?

RESPONSE: Thank you for pointing out this issue. Appropriate statistical analysis was performed and included in Figure S4.

3. Molecular dynamics (MD) simulation... This is too long paragraph or section. Authors need to slowly walk the readers with their thought by dividing this section into 3-4 paragraphs. Please be crisp in writing and to the point.

RESPONSE: We thank the reviewer for the excellent suggestion, which has improved the readability of our "Simulation section". We addressed this point by splitting up this section and also by making it more succinct and clearer. Part of the description

has been moved to the Method section.

4. Differential cellular internalization of the Pd nanocrystals: This reviewer would like to see measurements of ROS in explaining the differences in toxicity. It is okay to describe the results using penetration through membrane, but ROS generated in two facets must differ in giving different antibacterial properties. This will truly strengthen the manuscript. Authors should measure superoxide and hydroxyl radical concentrations in two kinds of facets under antibacterial study of Gram-positive and Gram-negative bacteria. Authors may like to consult the following references on the mechanism part of nanoparticles in general to get across the message out from this very nice piece of work: *Advances Colloid Interface Science* 2015, 225, 229-240; *Advances Colloid Interface Science* 2011, 166, 119-135.

RESPONSE: Thank you for this suggestion. We have evaluated the intracellular ROS levels by using CM-DCF fluorescence and representative fluorescent images, which are now included in Figure S7 and S9b. The results indicate that the antibacterial activity of the Pd nanocrystals is in agreement with the respective ROS levels generated. Furthermore, we used hydroethidine (HE) and 3'-p-(hydroxyphenyl) fluorescein (HPF) as fluorescent probes to respectively measure the formation of superoxide and hydroxyl radicals inside the bacterial cells. Relative to the control group, a strong fluorescence signal was observed upon addition of Pd nanocrystals, indicating the generation of hydroxyl and superoxide radicals (Figure S8).

Authors may like to change the title as Differential Pd-Nanocrystal Facets: Distinct Antibacterial Activity against Gram-positive and Gram-negative Bacteria.

RESPONSE: This point is well taken. We have followed the suggestion and made the corresponding change.

Reviewer #4 (Remarks to the Author):

The major claims presented in this manuscript are that palladium nanoparticles with

different facets, i.e. cubes or octahedrons induce oxidase and peroxidase activity on gram-positive and gram negative bacteria and in doing behave as antibacterial agents. The work includes experimental and theoretical studies to demonstrate the effect of the different facets on the bacteria. In general, the work is interesting, though it does not rise to the level to warrant the novelty or urgency for publication in Nature Communications.

RESPONSE: Thank you for your constructive comments, which have improved our manuscript (details below).

1. The claims of different facets having different effects on gram-positive vs. gram-negative bacteria, while interesting, is quite a stretch when only one species is used, and no experiments are done on other strains. Moreover, the specific strain used for each species is not referred to in the manuscript.

RESPONSE: We thank the reviewer for this valuable suggestion. We have added information about the specific strain used for each species in the Methods section. In order to further strengthen our conclusion, we investigated other bacterial species, as suggested by the referee, including *Enterococcus faecalis* (CICC 23658, gram-positive) and *Salmonella enteritidis* (CICC 21482, gram-negative). The antibacterial activities of the Pd nanocrystals against these additional bacterial species were evaluated as previously described. The results from these systems display similar trends regarding the antibacterial properties of the Pd nanocrystals as those observed for the gram-positive *S. aureus* and the gram-negative *E.coli* bacteria; these results are now included as Figure S12.

2. Another important aspect that should be taken into consideration is not only the type of surface but how the stabilizers are organized on the surface. The authors indicate that the stabilizer is PVP, but the role of PVP in these studies has not been considered. Stabilizers play a very important role in how they nanoparticles interact with bacteria and must be taken into consideration. In addition, including the role of the stabilizer in the computational studies would be important is getting a clear picture of the interactions that are being claimed.

RESPONSE: Thanks for your concern. This point was also raised by Referee #2 (see above). To address this issue, we utilized X-ray photoelectron spectroscopy (XPS) and our analysis (Figure S1d&e) demonstrate that the surface of the Pd nanocrystals did not get oxidized (no formation of Pd⁰ species). Also we observed the occurrence of charge transfer from either O atoms or N atoms from the PVP molecules to the Pd nanocrystals. Therefore, PVP is chemisorbed on the surface of the Pd nanocrystals by their O atoms and/or N atoms, where their molecular structure is well maintained, which is in agreement with previous report.¹ Moreover, to examine the role of PVP, we evaluated its impact on antibacterial activity. From our viability assays we observed that PVP, even at concentrations up to 600 µg/ml (Figure R1), showed no obvious antibacterial activity. Therefore, we think that the role of PVP in influencing the antibacterial activity of the Pd nanocrystals is (if any) negligible.

Figure R1 The effects of PVP on the antibacterial activity of Pd nanocrystals. Survival rates of *S. aureus* (a) and *E. coli* (b) after treatment with various concentrations of PVP.

Reference:

[1] Xian J, Hua Q, Jiang Z, et al. Size-dependent interaction of the poly (N-vinyl-2-pyrrolidone) capping ligand with Pd nanocrystals. *Langmuir*, 2012, 28(17): 6736-6741.

3. It is also important to note that several metal nanoparticles when present in media can slowly undergo dissolution or other transformations. There is no indication in this manuscript that this was a test conducted. It may be possible that one facet undergoes

dissolution differently than another resulting in the formation of ions that will cause toxicity toward the bacteria. Without such tests, the full picture of what may be taking place is missing.

RESPONSE: Thank you for this concern. We have evaluated the release of Pd²⁺ from the different Pd nanocrystals and the experimental details are included in the SI. According to the results (Figure S5), extremely small amounts of Pd²⁺ are released from the Pd nanocrystals. Moreover, morphological characterization of the Pd nanocrystals inside and outside of the bacteria, showed that no changes are observed for any of the Pd nanocrystals, which is an indication of their robust stability under these conditions (Figure S6). Thus, we suggested that even though there is release of trace amounts of Pd²⁺ from the Pd nanocrystals, their respective geometric characteristics remain robust under intracellular and extracellular conditions.

REVIEWERS' COMMENTS:

Reviewer #1 (Remarks to the Author):

In reviewing the revised manuscript, I feel that all the comments have been well addressed by the authors. I thus recommend its publication in Nature Commun as is.

Reviewer #2 (Remarks to the Author):

I have read the answer to my previous comments and I feel satisfied with them, I also read the answer to the other reviewers and I think that these comments are adequate. The corrected manuscript is well written and completed, now is ready for publication.

Reviewer #3 (Remarks to the Author):

As stated earlier in review of original submission that this is high quality manuscript and deserve to be published in Nature Communications.

This reviewer thoroughly looked the comments and is pleased that authors performed additional experiments to address fully the comments. truly, this has strengthen the manuscript. Authors have also addressed additional comments with strong and positive response.

This reviewer now recommends the publication without any further comments.

Reviewer #4 (Remarks to the Author):

The responses to the reviewer comments and the modifications made are adequate. I support its publication